# A conserved LDL-receptor motif regulates corin and CD320 membrane targeting in polarized renal epithelial cells

Ce Zhang[1], Yue Chen[1], Shijin Sun[1,2], Yikai Zhang[1,2], Lina Wang[1], Zhipu Luo[3], Meng Liu[1], Liang Dong[1], Ningzheng Dong[1,2]*, Qingyu Wu[1,4]*

[1]Cyrus Tang Hematology Center, Collaborative Innovation Center of Hematology, State Key Laboratory of Radiation Medicine and Prevention, Soochow University, Suzhou, China; [2]MOH Key Laboratory of Thrombosis and Hemostasis, Jiangsu Institute of Hematology, the First Affiliated Hospital of Soochow University, Suzhou, China; [3]Institute of Molecular Enzymology, Soochow University, Suzhou, China; [4]Cardiovascular & Metabolic Sciences, Lerner Research Institute, Cleveland Clinic, Cleveland, United States

**Abstract** Selective protein distribution on distinct plasma membranes is important for epithelial cell function. To date, how proteins are directed to specific epithelial cell surface is not fully understood. Here we report a conserved DSSDE motif in LDL-receptor (LDLR) modules of corin (a transmembrane serine protease) and CD320 (a receptor for vitamin B12 uptake), which regulates apical membrane targeting in renal epithelial cells. Altering this motif prevents specific apical corin and CD320 expression in polarized Madin–Darby canine kidney (MDCK) cells. Mechanistic studies indicate that this DSSDE motif participates in a Rab11a-dependent mechanism that specifies apical sorting. In MDCK cells, inhibition of Rab11a, but not Rab11b, expression leads to corin and CD320 expression on both apical and basolateral membranes. Together, our results reveal a novel molecular recognition mechanism that regulates LDLR module-containing proteins in their specific apical expression in polarized renal epithelial cells.

*For correspondence:
ningzhengdong@suda.edu.cn (ND);
wuq@ccf.org (QW)

Competing interests: The authors declare that no competing interests exist.

## Introduction

Renal reabsorption is of fundamental importance in body fluid, electrolyte, and metabolic homeostasis (*Nawata and Pannabecker, 2018*; *Polesel and Hall, 2019*; *van der Wijst et al., 2019*). The majority of water and solutes, including glucose, ions, amino acids, and vitamins, in the glomerular filtrate are reabsorbed in the proximal tubule, which consists of a single layer of polarized cuboidal epithelial cells. In these epithelial cells, receptors, transporters, and enzymes are located on distinct apical and/or basolateral membranes to carry out their functions. Under pathological conditions, altered tubular epithelial membrane polarity and cell surface protein distribution impair renal function and physiological homeostasis (*De Matteis and Luini, 2011*; *Rossier et al., 2002*; *Stoops and Caplan, 2014*; *Wilson, 2011*). To date, molecular and cellular mechanisms directing proteins to specific membranes in polarized renal epithelial cells are not fully understood.

Corin is a serine protease primarily expressed in the heart where it activates atrial natriuretic peptide (ANP), a cardiac hormone that regulates salt-water balance and blood pressure (*Li et al., 2017*; *Yan et al., 2000*). In addition to the heart, corin is also expressed in other tissues such as the uterus and kidney (*Cui et al., 2012*; *Ichiki et al., 2011*; *Wang et al., 2020*; *Yan et al., 1999*). In the kidney, corin expression is mostly restricted to tubular epithelial cells (*Dong et al., 2016*), suggesting a potential role in regulating sodium excretion and reabsorption. In supporting this hypothesis, reduced renal corin expression was found in rat kidney disease models with sodium retention and

proteinuria (*Polzin et al., 2010*). Similarly, reduced renal corin expression was reported in patients with chronic kidney disease (*Fang et al., 2013*).

Corin is a type II transmembrane protein with a cytoplasmic tail and an extended extracellular region containing distinct modular structures (*Hooper et al., 2000*; *Yan et al., 2000*). In non-polarized cardiomyocytes and human embryonic kidney (HEK) 293 cells, corin is expressed on the entire cell surface (*Gladysheva et al., 2008*; *Zhang et al., 2015*; *Zhang et al., 2017*). In contrast, corin is localized on the apical, but not basolateral, membrane in polarized renal epithelial cells (*Dong et al., 2016*; *Polzin et al., 2010*). Possibly, corin structural elements such as particular amino acids and/or carbohydrates are involved in a mechanism that controls distinct membrane targeting in polarized epithelial cells (*Mellman and Nelson, 2008*; *Overeem et al., 2015*; *Weisz and Rodriguez-Boulan, 2009*).

To test this hypothesis, we analyzed corin wild-type (WT) and mutants lacking individual modules in Madin–Darby canine kidney (MDCK) cells, a common model for polarized epithelial cells (*Cereijido et al., 1978*; *Rodriguez Boulan and Sabatini, 1978*). We conducted site-directed mutagenesis to identify individual amino acids crucial for apical corin expression. We verified our findings in other renal epithelial surface proteins and examined underlying molecular mechanisms. These experiments led us to identify a conserved motif in LDL-receptor (LDLR) class A modules (referred to as LDLR modules thereafter) that participates in the apical trafficking of corin and CD320 (a receptor for vitamin B12 uptake) in polarized renal epithelial cells.

## Results

### Apical corin expression in MDCK cells

Previously, corin expression on apical epithelial membranes was detected in rat and human kidneys (*Dong et al., 2016*; *Polzin et al., 2010*). To study corin expression on polarized epithelial membranes, we expressed human corin in MDCK cells. Consistently, confocal microscopy detected apical, but not basolateral, corin expression in polarized MDCK cells expressing ZO-1 protein at the apicolateral tight junction (*Figure 1A*). In contrast, corin expression in non-polarized HL-1 cardiomyocytes and HEK293 cells was on the entire cell membrane (*Figure 1A*).

Corin is synthesized as a zymogen, which is activated by proprotein convertase subtilisin/kexin-6 (PCSK6) on the cell surface (*Chen et al., 2015*). We analyzed corin activation in MDCK and control HEK293 cells. In western blotting, a similar corin fragment from the activation cleavage was observed in transfected MDCK and HEK293 cells (*Figure 1—figure supplement 1*). If these cells were treated with trypsin to remove surface proteins before being lysed, the activation cleavage fragment was not detected (*Figure 1—figure supplement 1*), indicating that, like in HEK293 cells, corin is activated on the surface of polarized MDCK cells.

To understand if the observed apical corin expression in polarized epithelial cells is a general characteristic of type II transmembrane serine proteases that function in diverse tissues to regulate physiological processes (*Antalis et al., 2011*; *Bugge et al., 2009*), we expressed a selected set of type II transmembrane serine proteases (hepsin, matriptase, and TMPRSS3-5) in MDCK cells. Neprilysin, a metalloproteinase known for its apical expression in renal epithelial and MDCK cells (*Dong et al., 2016*; *Jalal et al., 1991*), was included as a control. Hepsin, matriptase, and TMPRSS3-5 exhibited an apical and basolateral pattern that differed from apical corin expression (*Figure 1B*), indicating that specific apical expression in polarized epithelial cells is not a common feature among type II transmembrane serine proteases.

### Corin protein modules for specific apical expression in MDCK cells

To examine if determinant(s) for specific apical corin expression are located intracellularly or extracellularly, we analyzed corin mutants, in which the cytoplasmic tail (ΔCT) or extracellular non-protease modules (ΔFz1-SR) were deleted (*Figure 2A*). In MDCK cells, corin WT and the ΔCT mutant were expressed apically, whereas the ΔFz1-SR mutant was expressed apically and basolaterally (*Figure 2A*, *Figure 2—figure supplement 1A*), indicating that the determinant(s) for the apical-specific expression reside in the extracellular region between frizzled 1 (Fz1) and the scavenger receptor (SR) domains.

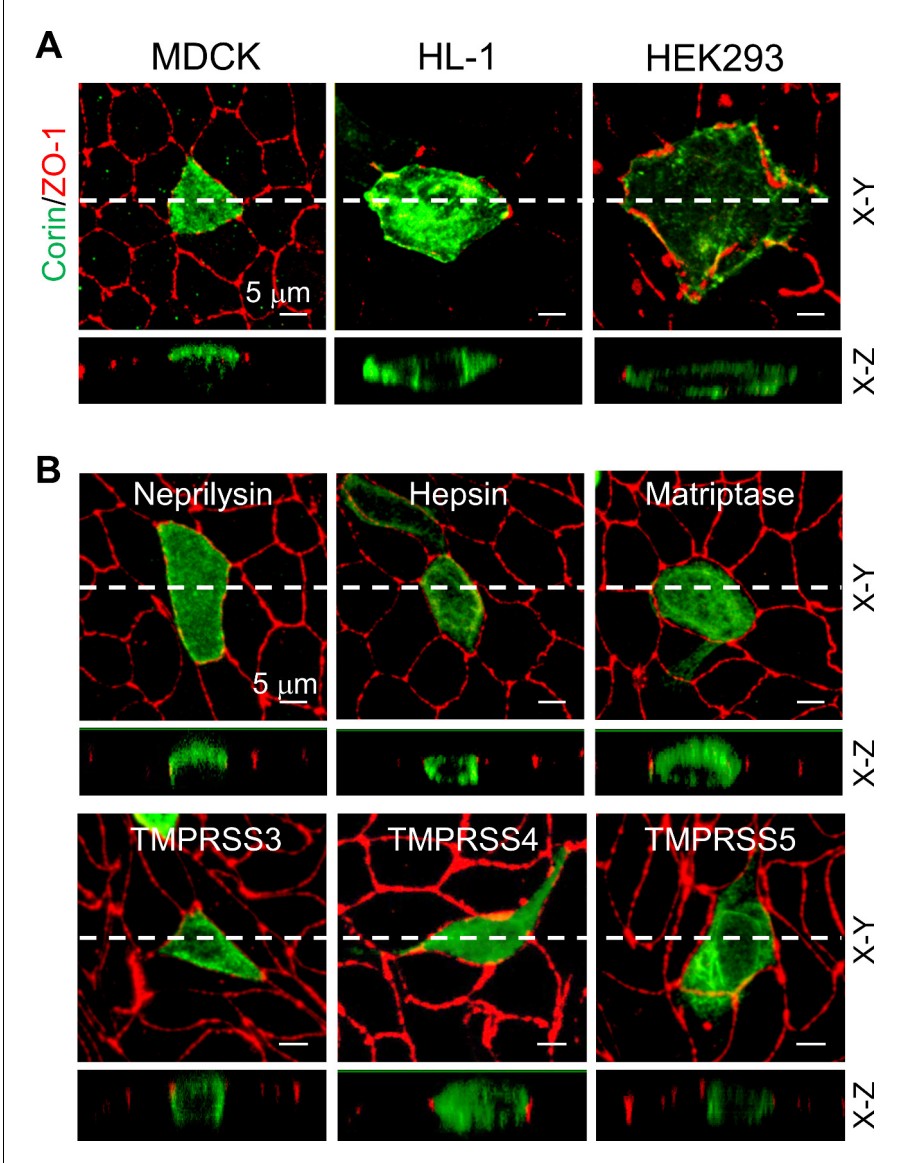

**Figure 1.** Expression of corin and other selected proteases in MDCK, HL-1, and HEK293 cells. (**A**) MDCK (left), HL-1 (middle), and HEK293 (right) cells were transfected with a plasmid expressing corin. After 72 hr, immunostaining was done to examine corin (green) and ZO-1 (red) (an indicator of the apicolateral tight junction) expression with confocal microscopy. X-Y and X-Z views are shown in top and lower panels, respectively. (**B**) Neprilysin, hepsin, matriptase, and TMPRSS3-5 (green) were expressed in transfected MDCK cells. Immunostaining and confocal microscopy were used to examine protein expression on apical and basolateral membranes. X-Y and X-Z views are shown in top and lower panels, respectively. Each image represents the data from five experiments. Scale bars: 5 μm.

The online version of this article includes the following figure supplement(s) for figure 1:

**Figure supplement 1.** Western blotting analysis of corin expression in HEK293 and MDCK cells.

We tested additional corin mutants, in which Fz and LDLR modules were deleted, individually or in combination (*Figure 2B*). In MDCK cells, the corin mutants lacking Fz1-2 (ΔFz1 and ΔFz2), LDLR1-5 (ΔR1-4 and ΔR45), or LDLR6-7 (ΔR67) modules were all expressed apically (*Figure 2B*). In contrast, the mutants, in which LDLR8 module was deleted either alone (ΔR8) or together with LDLR6 and/or

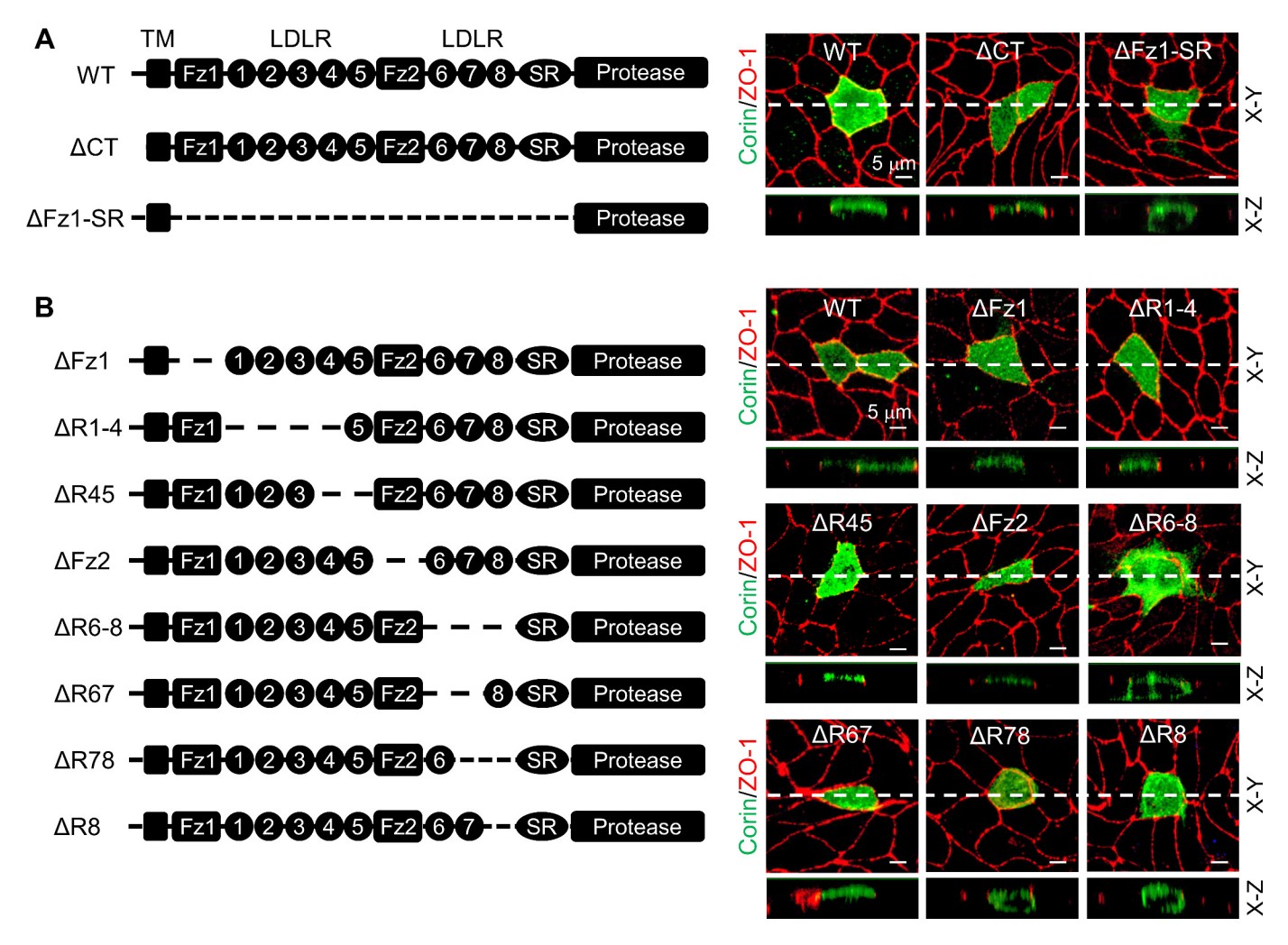

**Figure 2.** Analysis of corin deletion mutants in MDCK cells. (**A**) Human corin consists of an N-terminal cytoplasmic tail (CT), a transmembrane domain (TM) and an extracellular region with two frizzled (Fz) modules, eight LDLR modules, a scavenger receptor (SR) module, and a protease domain module. Corin wild-type (WT) and mutants lacking the CT (ΔCT) or Fz1-SR modules (ΔFz1-SR) were expressed in MDCK cells. Corin expression (green) on apical and basolateral membranes was examined by immunostaining and confocal microscopy with X-Y (top panels) and X-Z (lower panels) views. (**B**) Corin mutants lacking Fz and LDLR modules, individually or in combination, were expressed in MDCK cells and examined by immunostaining and confocal microscopy with X-Y (top panels) and X-Z (lower panels) views. Each image represents the data from five experiments. Scale bars: 5 μm.
The online version of this article includes the following source data and figure supplement(s) for figure 2:

**Figure supplement 1.** Quantitative analysis of corin expression on apical and basolateral membranes in MDCK cells.
**Figure supplement 1—source data 1.** Source data for *Figure 2—figure supplement 1A*.
**Figure supplement 1—source data 2.** Source data for *Figure 2—figure supplement 1B*.
**Figure supplement 2.** Analysis of corin mutants lacking individual N-glycosylation sites.
**Figure supplement 2—source data 1.** Source data for *Figure 2—figure supplement 2C*.

LDLR7 modules (ΔR6-8 and ΔR78), were expressed apically and basolaterally (*Figure 2B*). The results were confirmed by quantitative analysis of fluorescent signals on apical and basolateral membranes (*Figure 2—figure supplement 1B*). These results indicate that elements in or near corin LDLR8 module are important for specific apical expression.

N-glycans are known to mediate apical expression in polarized epithelial cells (*Potter et al., 2006*; *Weisz and Rodriguez-Boulan, 2009*). Human corin contains 19 N-glycosylation sites, some of

which, for example N761 and N1022, are required for corin expression on the surface of cardiomyo-cytes and HEK293 cells (*Wang et al., 2018*; *Wang et al., 2015*). We examined the corin mutants, in which N-glycosylation sites at N567 (N567Q), N651 (N651Q), N697 (N697Q), and N1022 (N1022Q) in or near LDLR6-8, SR and protease domain modules were abolished individually (*Figure 2—figure supplement 2A*). We found that all of the N-glycosylation site mutants had specific apical expression in MDCK cells (*Figure 2—figure supplement 2B,C*), indicating that the apical corin expression in polarized epithelial cells does not depend on individual N-glycans near LDLR8 module.

## Corin LDLR8 module residues in specific apical expression

Corin LDLR1-8 modules have a high degree of sequence similarities; yet only the deletion of LDLR8 module altered the specific apical expression of corin. We aligned corin LDLR1-8 module sequences and identified 11 residues in LDLR8 that were unique at corresponding positions (*Figure 3—figure supplement 1A*). Among them, five were conserved in vertebrate species from zebrafish to monkeys (*Figure 3—figure supplement 1B*). To test the functional importance of these residues, we replaced them with corresponding residues in LDLR6 module of the mutant ΔR78 that expressed both apically and basolaterally (*Figure 3A*). We found that Asp608 to Ser substitution (D608S), but not substitutions of the other four residues, restored the specific apical expression (*Figure 3A*, *Figure 3—figure*

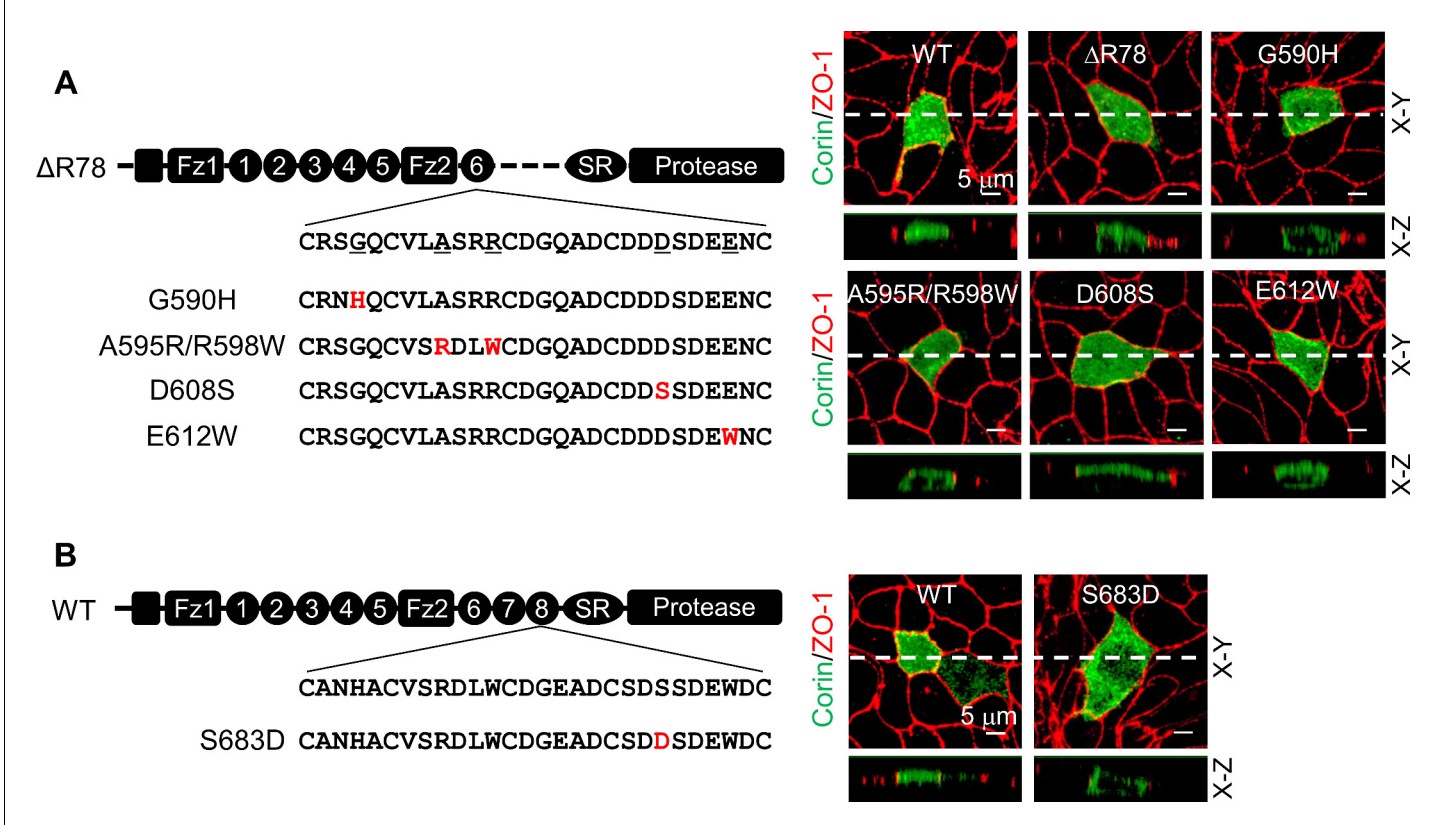

**Figure 3.** Expression of corin mutants with point mutations in LDLR6 module and without LDLR7-8 modules. (**A**) Corin wild-type (WT) and mutants lacking LDLR7-8 module without (ΔR78) or with point mutations at indicated positions (G590H, A595R/R598W, D608S, and E612W), indicated in red, were expressed in MDCK cells. Corin expression (green) on apical and basolateral membranes was examined by immunostaining and confocal microscopy with X-Y (top panels) and X-Z (lower panels) views. ZO-1 protein (red) was used as an indicator. (**B**) Corin WT and the mutant S683D expressed in MDCK cells were examined by immunostaining and confocal microscopy with X-Y (top panels) and X-Z (lower panels) views. Each image represents the data from five experiments. Scale bars: 5 μm.

The online version of this article includes the following source data and figure supplement(s) for figure 3:

**Figure supplement 1.** Alignments of corin LDLR module sequences.

**Figure supplement 1—source data 1.** Source data for *Figure 3—figure supplement 1C*.

**Figure supplement 1—source data 2.** Source data for *Figure 3—figure supplement 1D*.

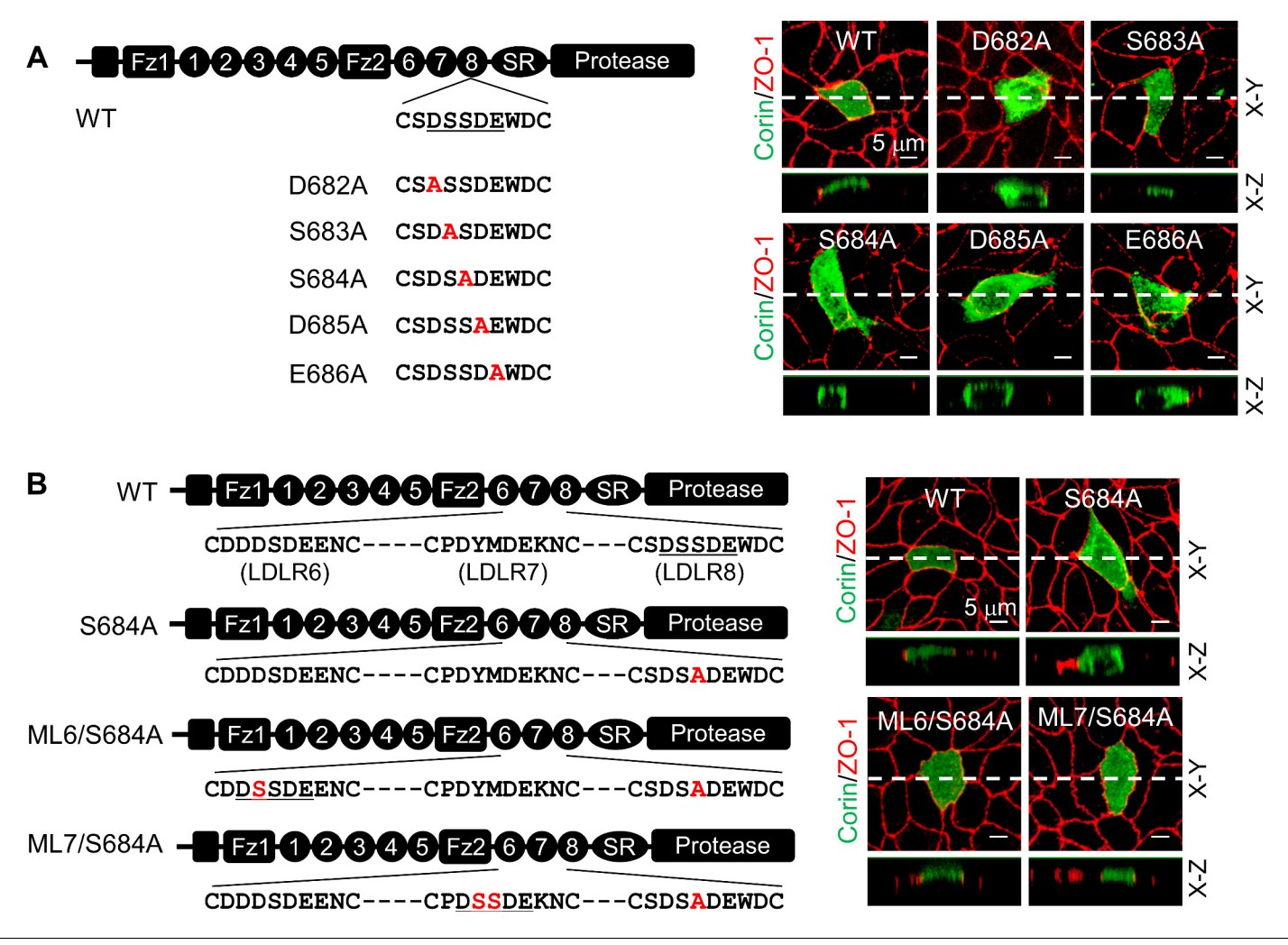

**Figure 4.** Expression of corin mutants with point mutations in the DSSDE motif in LDLR8 module. (**A**) Corin wild-type (WT) and mutants with point mutations at indicated positions (D682A, S683A, S684A, D685A, and E686A), indicated in red, were expressed in MDCK cells. Corin expression (green) on apical and basolateral membranes was examined by immunostaining and confocal microscopy with X-Y (top panels) and X-Z (lower panels) views. (**B**) Corin WT and S684A mutants without (S684A) or with the DSSDE motif created in LDLR6 (ML6/S684A) or LDLR7 (ML7/S684A) module expressed in MDCK cells were examined by immunostaining and confocal microscopy. X-Y (top panels) and X-Z (lower panels) views are indicated. Each image represents the data from five experiments. Scale bars: 5 μm.

The online version of this article includes the following source data and figure supplement(s) for figure 4:

**Figure supplement 1.** Analysis of the DSSDE motif in corin LDLR8 module.

**Figure supplement 1—source data 1.** Source data for *Figure 4—figure supplement 1B*.

**Figure supplement 1—source data 2.** Source data for *Figure 4—figure supplement 1C*.

**Figure supplement 2.** Western blotting of corin proteins in biotin-labeled cell membranes.

*supplement 1C*), suggesting that Ser683 in LDLR8 module is critical for specific apical corin expression in MDCK cells. To verify this result, we mutated Ser683 to Asp (the corresponding residue in LDLR6 module) in WT corin (*Figure 3B*). When expressed in MDCK cells, the S683D mutant was observed on both apical and basolateral membranes (*Figure 3B*, *Figure 3—figure supplement 1D*).

Ser683 is located in a DxSDE motif, where x is any amino acid, conserved among LDLR modules in many proteins (*Figure 4—figure supplement 1A*; *Herz, 2001*; *Mahley, 1988*). Within this motif, the last two residues (Asp-Glu) are part of a $Ca^{2+}$-binding cage that stabilizes the LDLR module

conformation (*Alam et al., 2016*; *Fass et al., 1997*; *Huang et al., 1999*; *North and Blacklow, 2000*). To understand the importance of the DSSDE motif in corin LDLR8 module, we mutated each residue in this motif to Ala individually (*Figure 4A*). In MDCK cells, D682A, S684A, D685A, and E686A mutants were expressed apically and basolaterally, whereas the S683A mutant was expressed apically, but not basolaterally (*Figure 4A*, *Figure 4—figure supplement 1B*). These results indicate that the DSSDE motif in corin LDLR8 module is important for specific apical expression and that within this motif Ser683 can be replaced by Ala without altering the membrane distribution pattern.

To verify these results, we tested corin mutants ML6/S684A and ML7/S684A, in which the DSSDE motif was introduced in the LDLR6 and LDLR7 modules, respectively, of the mutant S684A, which expressed apically and basolaterally (*Figure 4B*). In MDCK cells, both ML6/S684A and ML7/S684A mutants were expressed apically, but not basolaterally (*Figure 4B*, *Figure 4—figure supplement 1C*). In western blotting of biotin-labeled proteins, corin WT was found among apical, but not basolateral, membrane proteins, whereas the S684A mutant was among apical and basolateral membrane proteins (*Figure 4—figure supplement 2*). These results support the idea that the DSSDE motif is important for specific apical corin expression in polarized epithelial cells.

## Computational modeling of corin LDLR modules

To gain structural insights into the functional significance of the DxSDE motif in corin LDLR modules, we analyzed 15 LDLR module structures in the Protein Data Bank (*Figure 5—figure supplement*

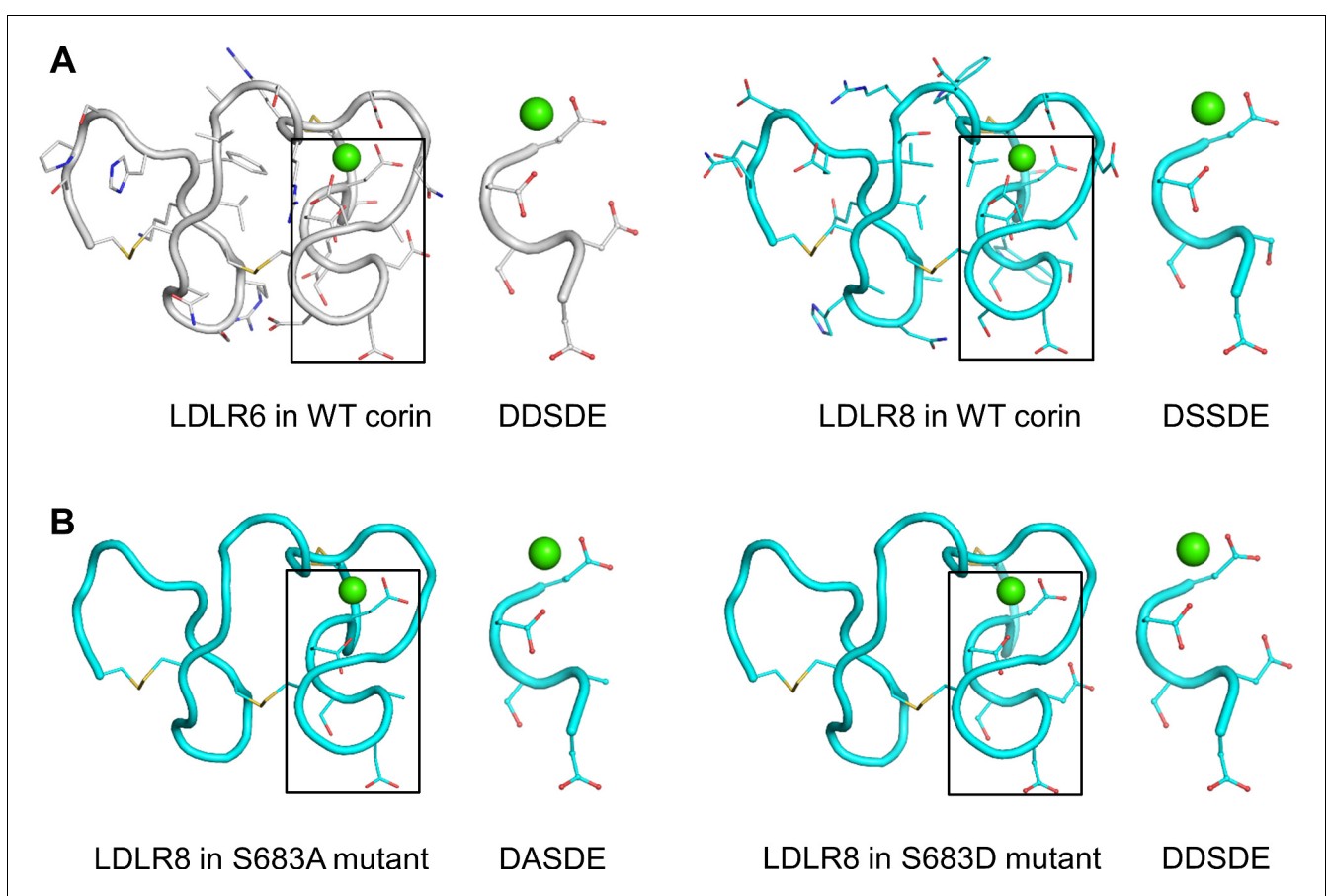

**Figure 5.** Structure models of corin LDLR6 and LDLR8 modules. (**A**) Structure models of human corin LDLR6 (left panel) and LDLR8 (right panel) modules, which were built based on the human VLDLR module four structure. (**B**) Structure models of corin S683A (left panel) and S683D (right panel) mutants. The conserved DxSDE motifs are indicated in boxes with enlarged images shown on the right side. The overall structures are shown in ribbon models. The side chains and disulfide bonds (yellow) are presented as sticks. The calcium ion is shown as a green sphere.

The online version of this article includes the following figure supplement(s) for figure 5:

**Figure supplement 1.** Structure alignments of LDLR modules.

1A; *Burley et al., 2019*), and built LDLR6 and LDLR8 module models based on the atomic coordinates of human VLDLR module 4 (*Banerjee et al., 2018*), which shared the highest degree of sequence identity and similarity with that of corin LDLR8 module (*Figure 5—figure supplement 1B*). Both corin LDLR6 and LDLR8 modules exhibited a similar overall topology with the Asp-Glu residues (the last two residues in the DxSDE motif) being part of the $Ca^{2+}$-binding cage (*Figure 5A*). In these models, Asp608 (second residue in the DDSDE motif) in LDLR6 module and Ser683 (second residue in the DSSDE motif) in LDLR8 module were located at a corner of the respective module and surface-exposed with potential to interact with other proteins (*Figure 5A*).

As described above, the mutant S683A was expressed apically (*Figure 4A*), whereas the mutant S683D was expressed apically and basolaterally (*Figure 3B*). In computational modeling, substitution of Ser683 to Ala (S683A) or Asp (S683D) did not disturb the overall structure of corin LDLR8 module (*Figure 5B*). The major difference was that the negatively-charged Asp683 side chain in the S683D mutant protruded more profoundly than the smaller and uncharged side chains of Ser683 and Ala683 did in corin WT and the mutant S863A, respectively (*Figure 5A,B*). These results suggest that a proper conformation in this corner region of LDLR modules may be important for protein interactions in specific membrane sorting.

## Apical expression of CD320 in renal epithelial cells

To understand if the DSSDE motif in corin LDLR8 module is present in other proteins expressed apically in polarized epithelial cells, we searched LDLR module-containing transmembrane proteins expressed in kidneys. We found that CD320, a type I transmembrane protein with two LDLR modules (*Alam et al., 2016*), has the DSSDE motif in the second LDLR module (*Figure 6A*). Structure alignment of CD320 LDLR1-2 and corin LDLR8 modules showed the DSSDE motif at the corner region of CD320 LDLR2 module (*Figure 6—figure supplement 1A–C*). By immunostaining of human

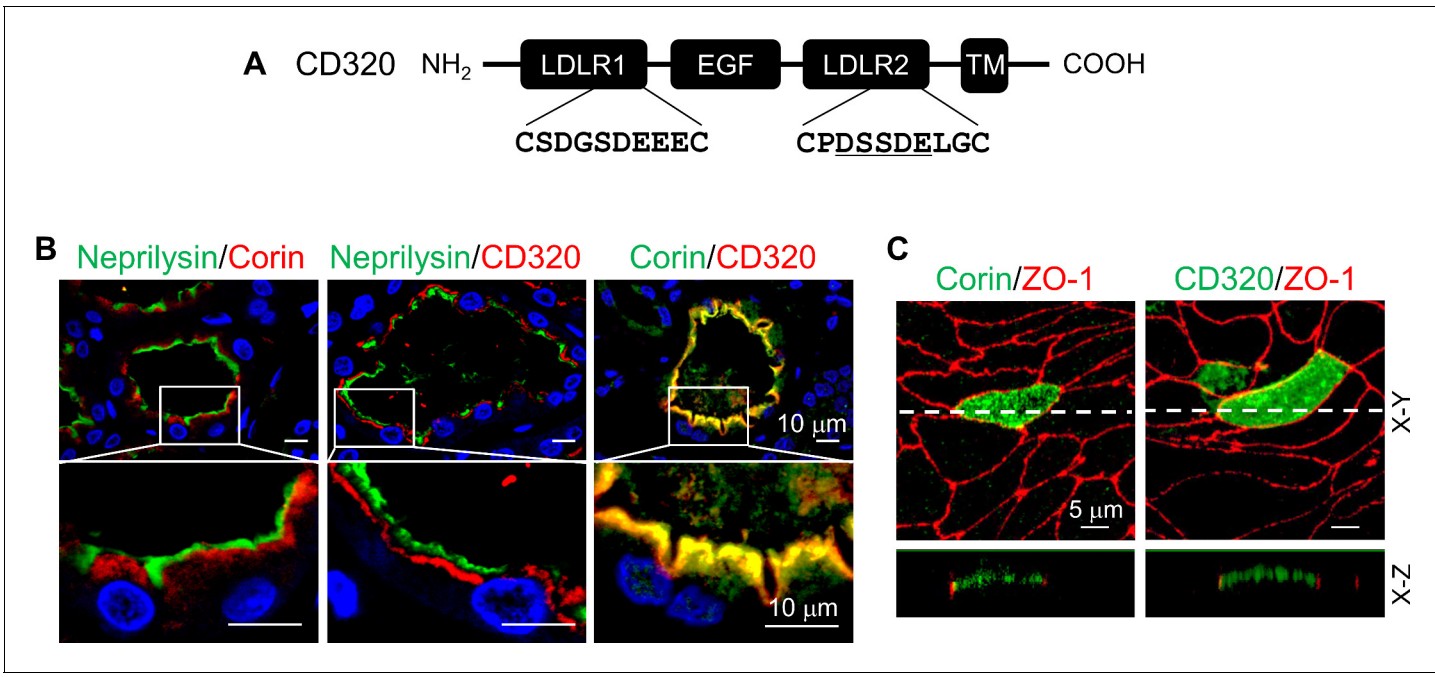

**Figure 6.** Expression of human CD320 in kidney sections and MDCK cells. (**A**) Human CD320 protein contains an N-terminal extracellular region with two LDLR modules and one EGF module, a transmembrane (TM) domain, and a C-terminal cytoplasmic tail. LDLR1 module has a DGSDE motif, whereas LDLR2 module has a DSSDE motif (underlined). (**B**) Co-staining of neprilysin (brush border marker) (green), corin (red or green) and CD320 (red) in proximal tubules in human kidney sections. Cell nuclei were stained in blue. Boxed areas in top panels are enlarged in lower panels. Scale bars: 10 μm. (**C**) Corin (green), CD320 (green), and ZO-1 (red) expression in transfected MDCK cells. X-Y and X-Z views are shown in top and lower panels, respectively. Scale bars: 5 μm. Each image represents data from five experiments.

The online version of this article includes the following figure supplement(s) for figure 6:

**Figure supplement 1.** Structural alignments of the LDLR domains of human CD320.

kidney sections, we found CD320 expression in the apical region underneath the brush border (neprilysin-positive) in proximal tubules, a pattern similar to that of corin (*Figure 6B*). Consistently, human CD320 was observed on the apical, but not basolateral, membrane in transfected MDCK cells (*Figure 6C*).

To test if the DSSDE motif in CD320 LDLR2 module regulates membrane targeting, we mutated residues in the DSSDE motif to Ala individually (*Figure 7A*). In MDCK cells, D160A, S162A, D163A, and E164A mutants were expressed apically and basolaterally, whereas the S161A mutant was expressed apically, but not basolaterally (*Figure 7A*, *Figure 7—figure supplement 1A*). These results are consistent with the findings in corin LDLR8 module mutants, indicating that the DSSDE motif in CD320 LDLR2 module also regulates membrane expression in polarized epithelial cells and that within this motif Ser161 can be replaced by Ala without noticeable effect on apical membrane targeting.

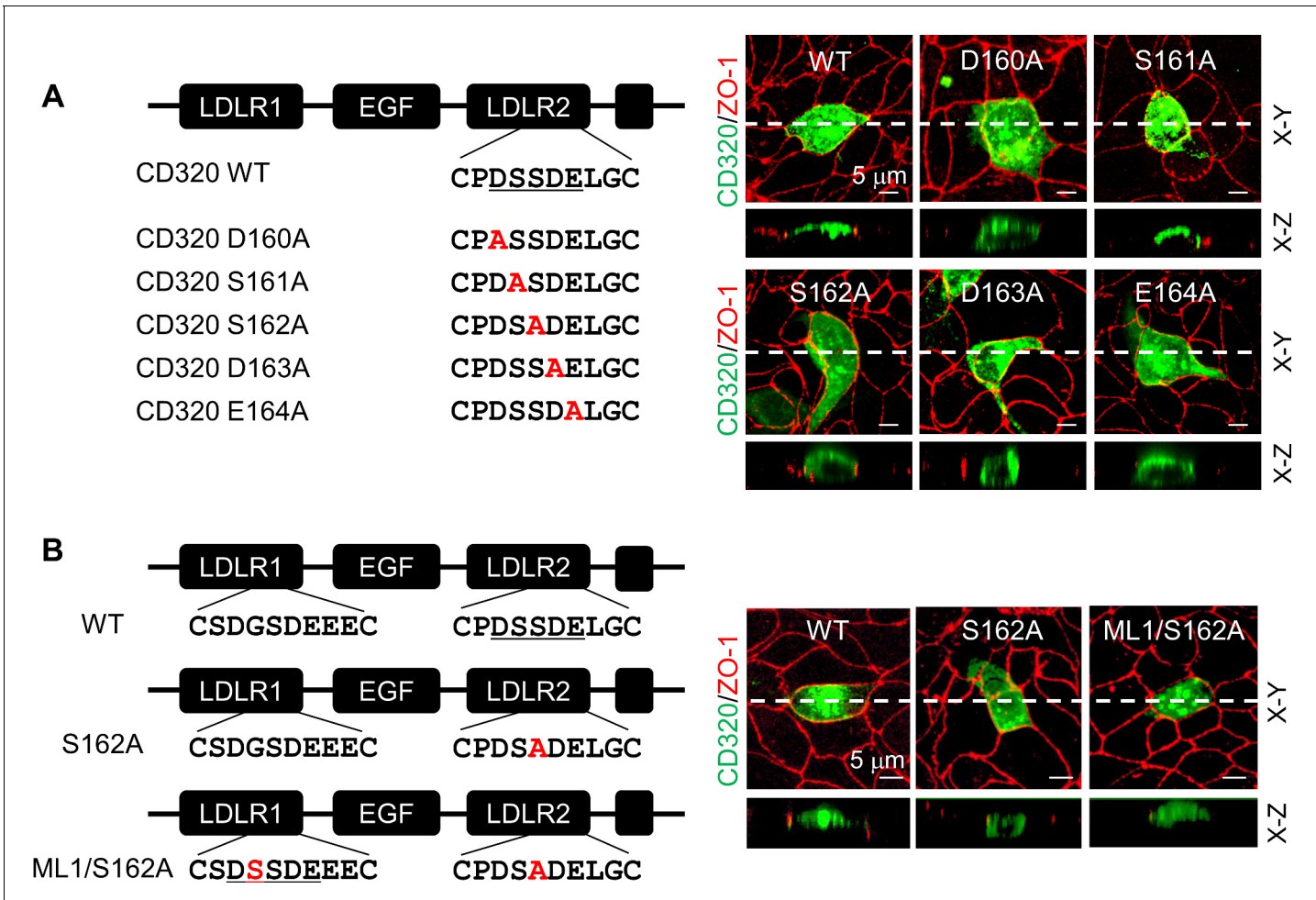

**Figure 7.** Expression of CD320 mutants with point mutations in the DSSDE motif in LDLR2 module. (**A**) Human CD320 WT and mutants with point mutations in LDLR2 module at indicated positions shown in red (D160A, S161A, S162A, D163A, and E164A) were expressed in MDCK cells. CD320 expression (green) on apical and basolateral membranes was examined by immunostaining and confocal microscopy with X-Y (top panels) and X-Z (lower panels) views. ZO-1 protein (red) was used as an indicator. (**B**) CD320 WT and S162A mutants without (S162A) or with the DSSDE motif created in LDLR1 module (ML1/S162A) expressed in MDCK cells were examined by immunostaining and confocal microscopy. X-Y (top panels) and X-Z (lower panels) views are indicated. Each image represents the data from five experiments. Scale bars: 5 µm.

The online version of this article includes the following source data and figure supplement(s) for figure 7:

**Figure supplement 1.** Quantitative analysis of CD320 and corin expression in MDCK cells.

**Figure supplement 1—source data 1.** Source data for *Figure 7—figure supplement 1A*.

**Figure supplement 1—source data 2.** Source data for *Figure 7—figure supplement 1B*.

Human CD320 LDLR1 module has a DGSDE, instead of DSSDE, motif (*Figure 7B*). The apical and basolateral expression of the CD320 mutant S162A indicated that the DGSDE motif did not support specific apical targeting. We tested the CD320 mutant ML1/S162A, in which the DSSDE motif was introduced in LDLR1 module of the mutant S162A (*Figure 7B*). In MDCK cells, the specific apical expression was restored in the ML1/S162A mutant (*Figure 7B*, *Figure 7—figure supplement 1B*), indicating that the DSSDE motif, whether in CD320 LDLR1 or LDLR2 module, can function similarly to support the specific apical targeting in polarized epithelial cells.

## Role of Rab11a and Rab11b in apical corin and CD320 expression

Rab11 proteins, that is Rab11a and Rab11b, play a key role in apical expression in polarized epithelial cells (*Kelly et al., 2012*; *Thuenauer et al., 2014*; *Welz et al., 2014*). To understand if Rab11 proteins are involved in apical corin and CD320 expression in MDCK cells, we co-expressed corin with human Rab11a, Rab11b, and dominant-negative (DN) mutants (Rab11aS25N and Rab11bS25N) that were inhibitory to Rab11a and Rab11b, respectively (*Thuenauer et al., 2014*; *Ying et al., 2016*). In cells expressing corin and recombinant Rab11a, the specific apical corin expression was not altered, whereas in cells expressing corin and DNRab11a, both apical and basolateral corin expression was observed (*Figure 8A*, *Figure 8—figure supplement 1A*). In contrast, apical, but not basolateral, corin expression was observed in cells co-expressing corin and recombinant Rab11b or DNRab11b (*Figure 8A*, *Figure 8—figure supplement 1A*), indicating that Rab11a, but not Rab11b, is involved in the specific apical corin expression in MDCK cells.

To verify these results, we made stable MDCK cells, in which Rab11a expression was down-regulated by shRNAs targeting the *RAB11A* gene (encoding Rab11a), as indicated by quantitative RT-PCR (*Figure 8B*) and western blotting (*Figure 8—figure supplement 1B*). In the *RAB11A*-knockdown cells, corin was expressed apically and basolaterally, whereas in control MDCK cells transfected with non-targeting shRNAs (shNC), corin was expressed apically (*Figure 8C*, *Figure 8—figure supplement 1C*). Similarly, CD320 was expressed apically in the control shRNA-transfected cells, but apically and basolaterally in the *RAB11A*-knockdown cells (*Figure 8C*, *Figure 8—figure supplement 1C*). In contrast, knocking down *RAB11B*, encoding Rab11b, in MDCK cells did not alter the apical expression of corin and CD320 (*Figure 8—figure supplement 2A–D*). These results suggest a possible cell membrane targeting mechanism, in which corin and CD320 are transported, likely via common recycling endosomes (CRE), to a Rab11a-positive apical recycling endosome (ARE), where LDLR modules with the DSSDE motif are recognized by a Rab11a-dependent mechanism that favors apical sorting (*Figure 8D*). For the corin and CD320 mutants, in which in the DSSDE motif is altered, such a recognition mechanism is eliminated, resulting in sorting to both apical and basolateral membranes (*Figure 8D*).

## Discussion

Selective protein distribution on distinct plasma membranes is of great important in epithelial biology (*Mostov et al., 2003*). In this study, we used MDCK cells as a model to examine corin expression on polarized epithelial membranes. In immunostaining, corin exhibited a specific apical expression pattern in MDCK cells, which differs from the overall membrane expression pattern in HL-1 cardiomyocytes and HEK293 cells. The distinct apical expression likely reflects the corin function in the lumen of renal tubules to regulate sodium excretion and reabsorption (*Dong et al., 2016*; *Li et al., 2017*).

Apical targeting in polarized epithelial cells has been studied extensively. To date, cytoplasmic tail, transmembrane domain, glycosyl-phosphatidylinositol (GPI) anchor, PDZ-binding motif, and N- or O-glycans have been identified as apical sorting signals (*Stoops and Caplan, 2014*; *Weisz and Rodriguez-Boulan, 2009*). Corin is a transmembrane, but not GPI-anchored, protein and contains no PDZ-binding motif or detectable amounts of O-glycans (*Liao et al., 2007*; *Yan et al., 2000*). In this study, we showed that the cytoplasmic tail and individual N-glycosylation sites near LDLR8 module were unnecessary for the specific apical expression of corin in MDCK cells. These results point to a yet unknown mechanism underlying apical corin expression in polarized renal epithelial cells.

By analyzing a series of corin mutants with modular deletions and point mutations, we identified a DSSDE motif in corin LDLR8 module required for specific apical expression in MDCK cells. In corin mutants, in which the DSSDE motif was disrupted in LDLR8 module, but re-introduced in LDLR6 or

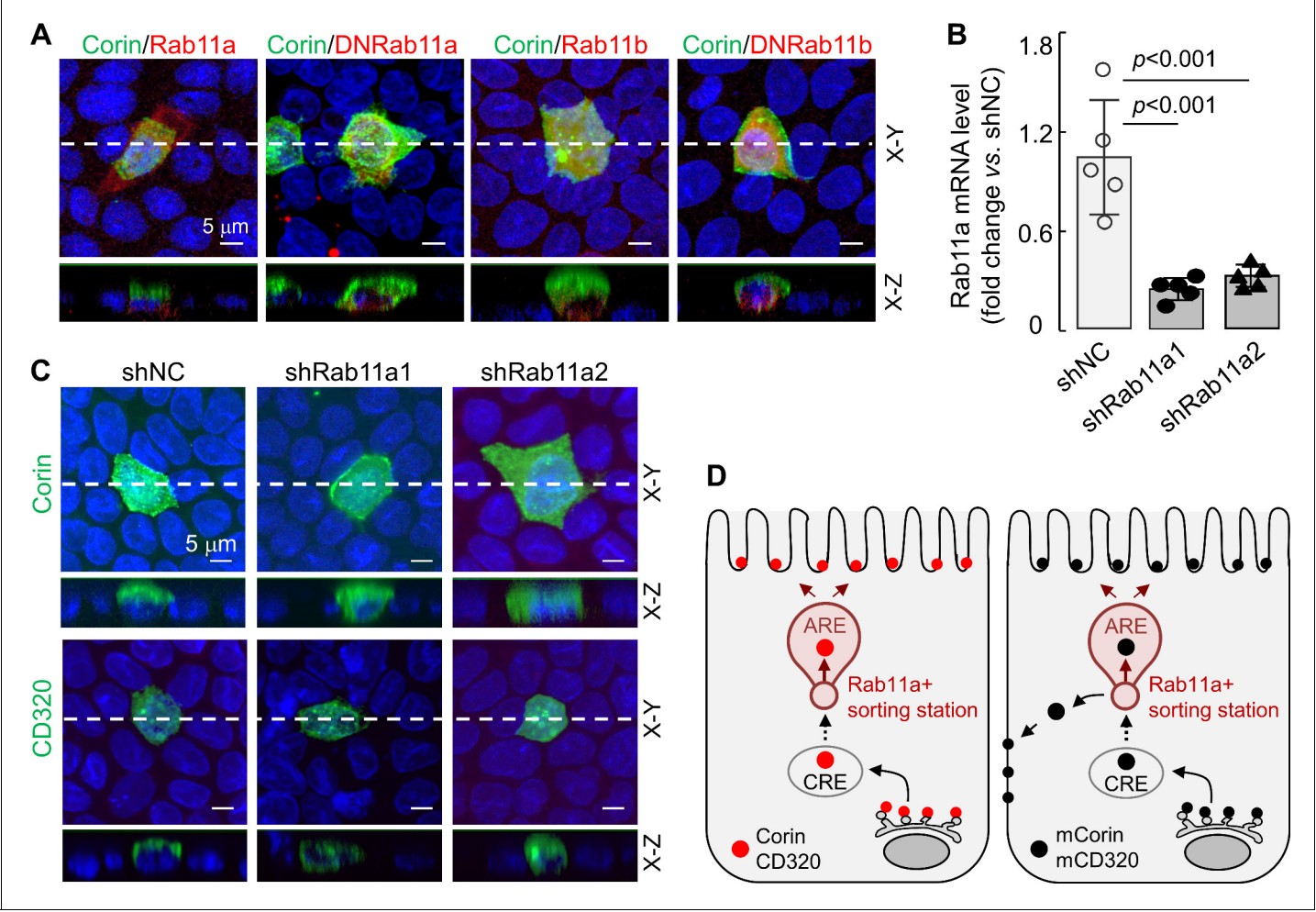

**Figure 8.** Effects of Rab11a inhibition or knockdown on corin and CD320 expression in MDCK cells. (**A**) Corin (green) and Rab11 proteins (red), including Rab11a, Rab11b, and dominant-negative (DN) mutants that were inhibitory to Rab11a and Rab11b, respectively, were co-expressed in MDCK cells. Immunostaining and confocal microscopy were used to analyze corin expression on apical and basolateral membranes, as indicated in X-Y and X-Z views. Each image represents the data from four experiments. Scale bars: 5 μm. (**B**) qRT-PCR analysis of Rab11a mRNA levels in MDCK cells transfected with two sets of shRNAs targeting the *RAB11A* gene (shRab11a1 and shRab11a2) or non-targeting control shRNAs (shNC). The data are mean ± SD from five experiments, analyzed by ANOVA. (**C**) Immunostaining of corin and CD320 in MDCK cells transfected with *RAB11A*-targeting shRNAs (shRab11a1 and shRab11b) or non-targeting shRNAs (shNC). X-Y and X-Z views are indicated. (**D**) A proposed model. Corin and CD320 (red dots) are transported from Golgi, likely via central recycling endosomes (CRE), to a Rab11a-positive sorting station, where the DSSDE motif is recognized for apical sorting via apical recycling endosomes (ARE) (left panel). Any pool of corin and CD320 that is endocytosed may also be re-sorted apically via Rab11a-positive endosomes. Mutations altering the DSSDE motif abolish such a mechanism, resulting in apical and basolateral expression of corin and CD320 mutants (right panel).

The online version of this article includes the following source data and figure supplement(s) for figure 8:

**Source data 1.** Source data for *Figure 8B*.
**Figure supplement 1.** Effects of Rab11a inhibition on apical corin and CD320 expression.
**Figure supplement 1—source data 1.** Source data for *Figure 8—figure supplement 1A*.
**Figure supplement 1—source data 2.** Source data for *Figure 8—figure supplement 1C*.
**Figure supplement 2.** Effects of Rab11b knockdown on corin and CD320 expression in MDCK cells.
**Figure supplement 2—source data 1.** Source data for *Figure 8—figure supplement 2A*.
**Figure supplement 2—source data 2.** Source data for *Figure 8—figure supplement 2D*.

LDLR7 module, the distinct apical expression of corin was restored. These results indicate that the DSSDE motif is a key element, transportable among LDLR modules, that specifies apical corin expression in polarized renal epithelial cells. Based on modeling analysis of known LDLR module structures (*Fass et al., 1997*; *Huang et al., 1999*; *North and Blacklow, 2000*), Ser683 (first Ser in the DSSDE motif) in corin is predicted to be surface exposed. In LDLR modules of the human patriarch LDLR, the corresponding surface area is a proposed ligand-binding site (*Fass et al., 1997*; *Huang et al., 1999*; *Russell et al., 1989*). Conceivably, the surface area around Ser683 in corin LDLR8 module may serve as a protein-interacting site for apical targeting in renal epithelial cells.

The finding of the DSSDE motif in corin LDLR8 module as a possible regulatory element in epithelial membrane targeting was unexpected. Corin protein has eight LDLR modules, six of which have the conserved DxSDE motif. Moreover, LDLR modules with the conserved DxSDE motif are present in numerous cell surface receptors and proteolytic enzymes (*Brown et al., 1997*; *Bugge et al., 2009*; *Herz, 2001*). We verified our findings in CD320, a receptor for transcobalamin in vitamin B12 uptake (*Bernard et al., 2018*; *Quadros and Sequeira, 2013*), which contains the DSSDE motif in its second LDLR module. CD320 expression in kidneys has been reported (*Arora et al., 2017*; *Park et al., 2009*), but its cellular and subcellular distribution in renal segments remained unclear. In immunostaining with human kidney sections, we found apical CD320 expression in proximal tubular epithelial cells. The result is consistent with CD320 function in vitamin B12 uptake, as most vitamin reabsorption occurs via cell surface receptors in renal proximal tubules (*Christensen and Birn, 1997*). Importantly, we found that the DSSDE motif in CD320 LDLR2 module was also required for specific apical expression in MDCK cells. These results suggest that the DSSDE motif may have a similar regulatory function in other LDLR module-containing proteins that are expressed apically in renal epithelial cells.

In polarized epithelial cells, Rab11a-positive endosomes are primary trafficking stations toward the apical membrane (*Hoekstra et al., 2004*; *Thuenauer et al., 2014*; *Wang et al., 2000*). In our study, overexpression of a dominant-negative Rab11a mutant (Rab11aS25N) or shRNA-mediated *RAB11A* knockdown in MDCK cells resulted in apical and basolateral expression of corin and CD320. In contrast, *RAB11B* knockdown did not alter the apical expression pattern of corin and CD320. It is possible that the DSSDE motif-containing LDLR modules in corin and CD320 are recognized by a Rab11a-dependent mechanism that specifies apical sorting. Mutations in the DSSDE motif abolish such a recognition mechanism, leading to apical and basolateral sorting of the mutant proteins. Further studies are required to understand the molecular basis for the potential Rab11a-dependent recognition mechanism.

In polarized epithelial cells, Rab11a is also known to mediate vesicle trafficking and recycling (*Perez Bay et al., 2016*; *Weisz and Rodriguez-Boulan, 2009*). At this time, it is unclear if corin undergoes endocytosis and recycling. In cardiomyocytes and HEK293 cells, corin is activated by PCSK6-mediated cleavage on the cell surface and subsequently undergoes ectodomain shedding (*Chen et al., 2015*; *Chen et al., 2018*; *Jiang et al., 2011*). In western blotting under reduction conditions, the corin protease domain fragment derived from activation cleavage was not detected intracellularly (*Chen et al., 2015*), suggesting that activated corin may not be internalized for recycling. In this study, we found similar corin activation on the surface of MDCK cells and did not detect the cleaved protease domain fragment inside the cells. Consistently, corin WT was detected among biotin-labeled apical, but not basolateral, membrane proteins. Additional studies with more direct and sensitive assays are required to determine if corin and CD320 undergo endocytosis and recycling in MDCK cells.

In summary, targeted apical expression is a key characteristic of polarized epithelial cells. Disturbed protein trafficking to distinct cell membranes in renal epithelial cells has been shown to cause kidney diseases. In this study, we have identified a conserved DSSDE motif in corin and CD320 LDLR modules as a regulatory element in apical sorting in MDCK cells. This regulatory function is likely mediated by a Rab11a-dependent mechanism. The DSSDE motif is present in other proteins with LDLR modules. Our findings should encourage more research to examine if analogous motifs in other LDLR module-containing proteins have a similar role in apical membrane targeting in polarized epithelial cells.

# Materials and methods

## Key resources table

| Reagent type (species) or resource | Designation | Source or reference | Identifiers | Additional information |
|---|---|---|---|---|
| Gene (*H. sapiens*) | CORIN | NCBI | NM_006587.4 | |
| Gene (*H. sapiens*) | ST14 (matriptase) | NCBI | NM_021978.4 | |
| Gene (*H. sapiens*) | TMPRSS3 | NCBI | NM_032404.3 | |
| Gene (*H. sapiens*) | TMPRSS4 | NCBI | NM_019894.4 | |
| Gene (*H. sapiens*) | TMPRSS5 | NCBI | NM_030770.4 | |
| Gene (*H. sapiens*) | MME (neprilysin) | NCBI | NM_007287.4 | |
| Gene (*H. sapiens*) | CD320 | NCBI | NM_016579.4 | |
| Cell line (*H. sapiens*) | HEK293 | ATCC | CRL-1573 | STR profiling, no mycoplasma contamination |
| Cell line (*Canis familiaris*) | MDCK | CCTCC* | 3142C0001000000147 | STR profiling, no mycoplasma contamination |
| Cell line (*Mus musculus*) | HL-1 | Sigma-Aldrich | SCC065 (Originally from William Claycomb) | No mycoplasma contamination |
| Transfected construct (*H. sapiens*) | Corin plasmid | PMID:14559895 | | pcDNA 3.1/V5-His |
| Transfected construct (*H. sapiens*) | Neprilysin plasmid | PMID:27343265 | | pCMV3-C-Flag |
| Transfected construct (*H. sapiens*) | Hepsin plasmid | PMID:31395734 | | pcDNA 3.1/V5-His |
| Transfected construct (*H. sapiens*) | Matriptase plasmid | This paper | | pcDNA 3.1/V5-His |
| Transfected construct (*H. sapiens*) | TMPRSS3 | This paper | | pcDNA 3.1/V5-His |
| Transfected construct (*H. sapiens*) | TMPRSS4 | This paper | | pcDNA 3.1/V5-His |
| Transfected construct (*H. sapiens*) | TMPRSS5 | This paper | | pcDNA 3.1/V5-His |
| Transfected construct (*H. sapiens*) | CD320 | This paper | | pcDNA 3.1/N-Flag |
| Transfected construct (*Canis familiaris*) | Rab11a | This paper | | pCMV3-C-OFPSpark |
| Transfected construct (*Canis familiaris*) | Rab11b | This paper | | pCMV3-C-OFPSpark |
| Transfected construct (*H. sapiens*) | Corin mutants | This paper | | pcDNA 3.1/V5-His |
| Transfected construct (*H. sapiens*) | CD320 mutants | This paper | | pcDNA 3.1/N-Flag |
| Antibody | Anti-corin (rabbit polyclonal) | PMID:27343265 | | 1:1000 |
| Antibody | Anti-corin (mouse monoclonal) | PMID:22437503 | | 1:1000 |

*Continued on next page*

*Continued*

| Reagent type (species) or resource | Designation | Source or reference | Identifiers | Additional information |
|---|---|---|---|---|
| Antibody | Anti-CD320 (rabbit polyclonal) | Atlas Antibodies | HPA073489 | 1:10 |
| Antibody | Anti-neprilysin (mouse monoclonal) | Boster | BM3411 | 1:100 |
| Antibody | Anti-Flag (mouse monoclonal) | Sigma | F1804 | 1:500 |
| Antibody | Anti-V5 (mouse monoclonal) | Thermo Fisher | R96025 | 1:500 |
| Antibody | Anti-V5-HRP (mouse monoclonal) | Thermo Fisher | R96125 | 1:5000 |
| Antibody | Anti-ZO-1 (rabbit polyclonal) | Cell Signaling | 8193 s | 1:300 |
| Antibody | Anti-IgG-Alexa-488 (donkey polyclonal) | Thermo Fisher | A32766 | 1:500 |
| Antibody | Anti-IgG-Alexa-594 (goat polyclonal) | Thermo Fisher | A32740 | 1:500 |

[*]CCTCC: China Center for Type Culture Collection.

## Human kidney samples

The study involving human tissues was approved by the Ethics Committee of Soochow University (2013052). Kidney tissues were obtained from patients who underwent nephrectomy for renal cancers at a university affiliated hospital. Informed consents were obtained from the donners. Tissues were dissected from non-tumor areas, fixed in 4% (v/v) formaldehyde, and embedded in paraffin. Sections were made, stained with hematoxylin and eosin (H and E), and verified by a pathologist for normal renal histology without fibrosis, hemorrhages, and necrosis. All procedures using tissue sections were carried out according to the approved protocol.

## Immunostaining in tissue sections

To examine corin and CD320 expression and subcellular localization in renal tubular epithelial cells, kidney tissues were fixed in 4% (v/v) formaldehyde, embedded in paraffin, cut in 3 μm sections, and mounted on chrome alum gelatin-coated slides (Aladdin, p1650). After dewaxing, sections were boiled in sodium citrate buffer (0.01 mM, pH 6.0) for 1 min for antigen retrieval, and soaked in a methanol solution containing 3% (v/v) hydrogen peroxide for 5 min at room temperature to block endogenous peroxidase activity. After incubation with 5% (w/v) bovine serum albumin (BSA) at 37°C for 30 min to reduce non-specific background, an antibody against corin (described above, 1:1000), CD320 (Atlas Antibodies, HPA073489, 1:10), or neprilysin (Boster, BM3411, 1:100) was added and incubated at 4°C overnight. After washing with PBS, secondary antibodies Alexa Fluro488-donkey anti-mouse IgG (Thermo Fisher, A32766, 1:500) and Alexa Fluro594-goat anti-rabbit IgG (Thermo Fisher, A32740, 1:500) were added and incubated at 37°C for 1 hr. After washing, the sections were mounted with a solution containing 4–6-diamidino-2-phenylindole (DAPI) (Southern Biotech, 0100–20). Cell images were examined and recorded using a confocal microscope (Olympus, FV1000).

## Cell culture

MDCK (China Center for Type Culture Collection, STR profiled) and HEK293 (ATCC, STR profiled) cells were grown in Dulbecco's modified Eagle's medium (DMEM) (Corning, 10-0130CVRC) with 10% fetal bovine serum (FBS) (Gibco, 16000–044) at 37°C with 5% $CO_2$. Murine HL-1 cardiomyocytes, originally from Dr. William Claycomb (*Claycomb et al., 1998*), were grown in Claycomb medium (Hyclone, 51800C) with 10% FBS and 4 mM/L L-glutamine (Sigma, 21051024) at 37°C with 5% $CO_2$.

## Plasmid constructs

The plasmids expressing human neprilysin, hepsin, and corin WT and mutants were described previously (*Dong et al., 2016*; *Knappe et al., 2004*; *Wang et al., 2019*). Human full-length cDNAs encoding matriptase, TMPRSS3, TMPRSS4, and TMPRSS5 were amplified from human cell line-derived cDNA libraries and inserted into pcDNA 3.1/V5-His plasmid (Thermo Fisher, K4800-01), which encodes a C-terminal V5 tag. Human CD320 cDNA (106–849 bp) from a HEK293 cell-derived library was cloned into pcDNA 3.1 plasmid, which had an inserted 5' sequence encoding the CD33 signal peptide and a Flag tag. Plasmids encoding human Rab11a and Rab11b with a C-terminal orange fluorescent protein (OFP) tag were made by cloning cDNAs from the HEK293 cell-derived library into pCMV3-C-OFPSpark (Sino Biological, China). Additional site-directed mutagenesis was done to make plasmids expressing mutant corin, CD320, Rab11a, and Rab11b proteins using ClonExpress One Step Cloning kit (Vazyme, China, C112). All plasmids used in this study were verified by DNA sequencing.

## Cell transfection and immunostaining

MDCK, HEK293, and HL-1 cells cultured in 12-well plates with glass coverslips (20 mm in diameter) were transfected with plasmids using PolyJet reagents (SignaGen Laboratories, SL100688) at 37°C. After 6 hr, the cells were switched to fresh medium. After 72 hr, the cells were fixed with pre-cooled methanol at room temperature for 5 min and incubated with 5% (w/v) BSA in PBS at 37°C for 1 hr. After washing with PBS, the cells were incubated with an anti-V5 or Flag antibody and an anti-ZO-1 antibody at 37°C for 1 hr, followed by incubation with Alexa Fluro-488 or 594-labeled secondary antibodies at 37°C for 1 hr. After washing, coverslips were mounted with a DAPI solution (Southern Biotech, 0100–20) to stain DNA in cell nuclei. The cells were examined with a confocal laser scanning microscope (Olympus, FV1000) along the X-Z axis from the bottom to the top. X-Y plane photos (8 µs/picture) were taken at each 0.45 µm and used to reconstitute X-Z axis images. Image J software was used to analyze fluorescent intensity on apical (guided by ZO-1 staining) and basolateral (total staining minus apical staining) membranes and calculate the ratio of $F_{BL}/F_{Total}$, where $F_{BL}$ is the fluorescent intensity on basolateral membranes and $F_{Total}$ is the total fluorescent intensity on apical and basolateral membranes. Values of black area without fluorescent staining were set as the background. No noise reduction of the image data was performed.

## Trypsin digestion of cell surface proteins and western blotting

To examine proteins on MDCK and HEK293 cell surface, transfected cells expressing corin were treated with 0.25% (w/v) trypsin and 0.02% EDTA (w/v) (Gibco, 25200) at 37°C for 30 s. DMEM with 10% FBS was added to neutralize trypsin activity. After washing with PBS, the cells were lysed in a buffer containing 1% (v/v) Triton X-100, 50 mM Tris–HCl (pH 8.0), 150 mM NaCl, and a protease inhibitor mixture (1:100, Roche Applied Science, 04693116001). Proteins were separated by SDS-PAGE under reducing conditions with 2.5% (v/v) β-mercaptoethanol in the Laemmli buffer (Bio-Rad) and transferred onto polyvinylidene difluoride membranes (Thermo Fisher) in an apparatus (Bio-Rad Trans-Blot, 300 mA) with 25 mM Tris base, 190 mM glycine, and 20% (v/v) methanol at 4°C for 60 min. Western blotting was done using a horseradish peroxidase (HRP)-conjugated anti-V5 antibody (1:5000, Thermo Fisher, R96125). After incubation with a chemiluminescent substrate (EZ-ECL) (Biological Industries, K-12045), western blots were exposed to X-ray films, which were analyzed by a scanner (CanoScan LiDE 110, Canon).

## Biotin-labeling of cell membrane proteins

MDCK cells transfected with a control vector or plasmids expressing corin WT and the S684A mutant were cultured with G418 (2 mg/mL) (Gibco) for 3–4 weeks to obtain stable cell clones. The selected cells were grown in Transwell culture plates (Costar, 3419) with 75 mm inserts containing polycarbonate membranes (pore size: 0.4 µm). When reaching confluency, the cells were washed with cold (4°C) PBS. Sulfo-NHS-biotin (0.25 mg/mL) (Thermo Fisher) was added to the upper or the bottom chamber to label proteins on apical and basolateral membranes, respectively. After 5 min at 4°C, the labeling reaction was stopped with a glycine solution (100 mM). The cells were washed and lysed, as described above. Biotin-labeled proteins were precipitated with immobilized NeutrAvidin beads (Thermo Fisher) at 4°C overnight and analyzed by SDS-PAGE and western blotting under reducing

conditions, as described above. Proteins in cell lysates were analyzed in parallel as controls together with an HRP-conjugated anti-GAPDH antibody (MultiScience, ab011-100, 1:10000).

## Gene knockdown and quantitative RT-PCR (qRT-PCR)

To knock down the **RAB11A** or **RAB11B** gene, MDCK cells were transfected with two sets of **RAB11A**-targeting (shRab11a1 and shRab11a2) or **RAB11B**-targeting (shRab11b1 and shRab11b2) shRNAs or non-targeting control shRNAs (shNC) (GeneParma, China). Stable cells were selected with puromycin (1 µg/mL) (Gibco). Total RNAs were isolated using the EZNA HP Total RNA kit (OMEGA) and reverse-transcribed to make cDNAs using the RevertAid First Strand cDNA Synthesis kit (Thermo Fisher). Rab11a and Rab11b mRNA levels were analyzed by qRT-PCR using SYBR Green MasterMix in the PRISM 7500 Sequence Detection System (Applied Biosystems). **The primers used for RAB11A** were 5'-ATT TGC GTC ATC TCA GGG CA-3' (sense) and 5'-GTT TCT GGG AAA CAA TGC GGT-3' (antisense) and for **RAB11B** were 5'-CTA TGG AGG GAA TGT CGC TAT C-3' (sense) and 5'-CTG TCT CTG GTG AGT CTT CTA G-3' (antisense). A control (18S mRNA) was analyzed in parallel. Western blotting was used to verify Rab11a and Rab11b protein levels in the targeted cells with primary antibodies against Rab11a (Cell Signaling, 2413, 1:1000) and Rab11b (Boster, M04526, 0.5 µg/mL).

## Modeling of LDLR module structures

The sequence of human corin LDLR8 module was subjected to the Protein Data Bank (PDB) (*Burley et al., 2019*) to identify homologous structures. Fifteen matched structures with low E-values and high resolution were chosen for further analysis. The best matched structure was that of module four in the human VLDLR (6BYV, residues 154–188) (*Banerjee et al., 2018*) with 51% (18/35 residues) identities and 69% (24/35 residues) similarities compared to the corin LDLR8 module sequence (residues 655–689). The VLDLR structure was used as a template to build homology models for corin LDLR6, LDLR8, and mutant LDLR8 (S683A and S683D) modules using SWISS-MODEL (*Waterhouse et al., 2018*). Figures of model structures and sequence alignments were generated by PyMoL software (Molecular Graphics System, V2.0, Schrödinger). CD320 LDLR1-2 module structures were extracted from the human transcobalamin and CD320 complex structure (PDB ID: 4ZRP) (*Alam et al., 2016*). The overall structure alignment was generated by PyMOL based on the Cα of all residues. The local alignment of CD320 LDLR2 and corin LDLR8 modules was generated by PyMOL based on the Cα of the DSSDE motif.

## Statistical analysis

Data were analyzed with Prism six software (Graphpad). Student's *t* test was used to analyze data from two groups and ANOVA followed by Tukey's multiple comparison test was used to analyze data from three or more groups. P values of < 0.05 were considered to be statistically significant.

# Acknowledgements

We thank Lin Qi and Boxing Xue for their assistance in immunostaining studies. This work was supported in part by grants from the National Natural Science Foundation of China (81873840 and 81570457), and the Priority Academic Program Development of Jiangsu Higher Education (to Soochow University).

# Additional information

### Funding

| Funder | Grant reference number | Author |
| --- | --- | --- |
| National Natural Science Foundation of China | 81873840 | Ningzheng Dong |
| National Natural Science Foundation of China | 81570457 | Ningzheng Dong |

The funders had no role in study design, data collection and interpretation, or the decision to submit the work for publication.

### Author contributions

Ce Zhang, Conceptualization, Data curation, Formal analysis, Validation, Investigation, Visualization, Methodology, Writing - original draft, Writing - review and editing; Yue Chen, Shijin Sun, Data curation, Investigation, Methodology, Writing - review and editing; Yikai Zhang, Data curation, Formal analysis, Investigation, Visualization, Writing - review and editing; Lina Wang, Investigation, Methodology, Writing - review and editing; Zhipu Luo, Data curation, Investigation, Visualization, Writing - original draft, Writing - review and editing; Meng Liu, Supervision, Investigation, Methodology, Project administration, Writing - review and editing; Liang Dong, Resources, Data curation, Investigation, Visualization, Writing - review and editing; Ningzheng Dong, Conceptualization, Resources, Formal analysis, Supervision, Funding acquisition, Validation, Writing - original draft, Project administration, Writing - review and editing; Qingyu Wu, Conceptualization, Formal analysis, Supervision, Investigation, Visualization, Writing - original draft, Writing - review and editing

### Author ORCIDs

Qingyu Wu https://orcid.org/0000-0003-0561-9315

### Ethics

Human subjects: The study involving human tissue samples was approved by the Ethics Committee of Soochow University (#2013052). Informed consents were obtained from the donors.

### Decision letter and Author response

Decision letter https://doi.org/10.7554/eLife.56059.sa1
Author response https://doi.org/10.7554/eLife.56059.sa2

## Additional files

### Supplementary files

• Transparent reporting form

### Data availability

All data generated or analysed during this study are included in the manuscript and supporting files.

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
