## [Decision Letter]

**Acceptance summary:**

A major pursuit in the past three decades of polarized membrane traffic research has been to identify apical sorting determinants, with the ultimate goal of identifying the "sorting" machinery that recognizes these motifs. This study defines an extracellular domain motif that specifies the accumulation of corin (a serine proteinase) and CD320 (receptor for B12 uptake) at the apical region of polarized MDCK cells. These two proteins contain conserved LDL receptor (LDLR) class A modules in their extracellular domains. Mutation or removal of a DSSDE motif in a subset of these LDLR modules causes a partial redistribution of the proteins to the basolateral membrane whereas transplant of this motif to unsorted forms of the proteins causes their apical redistribution. Other studies demonstrate a role of Rab11a in the preferential apical sorting of the proteins. Thus, the DSSDE motif represents a "regulatory element" that controls the expression of LDLR module-containing proteins at the apical pole of kidney cells.

**Decision letter after peer review:**

Thank you for submitting your article "A conserved LDL receptor motif regulates corin and CD320 membrane targeting in polarized renal epithelial cells" for consideration by *eLife*. Your article has been reviewed by three peer reviewers, one of whom is a member of our Board of Reviewing Editors, and the evaluation has been overseen by Olga Boudker as the Senior Editor. The reviewers have opted to remain anonymous.

The reviewers have discussed the reviews with one another and the Reviewing Editor has drafted this decision to help you prepare a revised submission.

Summary:

The major contribution of this article is the potentially novel finding that the apical membrane proteins corin and CD320 contain a peptide-based, sorting determinant in their extracellular domain that specifies apical targeting in kidney cells. Extensive mutagenesis and MDCK expression experiments were carried out to characterize the DXSDE motif present in Corin, a transmembrane serine protease, and CD320, a receptor for Vitamin B12 uptake. Mutation or removal of this motif causes a partial redistribution of the proteins to the basolateral membrane whereas transplant of this motif to unsorted forms of the proteins causes their apical redistribution. Other studies demonstrate a role of Rab11a in the preferential apical sorting of the proteins. While we don't learn how the signal works- especially in relation to rab11a function- the finding is significant given its novelty and appearance in multiple proteins. However, the review has raised issues that must be satisfactorily addressed before the manuscript is considered acceptable for publication.

Essential revisions:

1) The apical sorting of the constructs is not entirely convincing. There are several issues. Foremost is the lack of quantification of the microscopy-based assay. The authors present single selected images, which is not satisfactory. Additionally, given the lack of resolution of the microscopy in the Z-axis, it is not entirely convincing that the signal being detected is surface-localized rather than, at least partly, intracellular. One reviewer states "the gold standard has been to use cell-surface biotinylation (or related techniques such as cell-surface immunoprecipitation). While transfection efficiency can affect one's ability to perform these studies, the standard for decades has been to make stable cell lines or use viral transduction approaches. These latter approaches could be done using a small subset of the mutants."

2) It is not convincing that the sorting involves direct apical delivery without endocytosis of a basolateral pool of protein. The possibility of endocytosis of a basolateral pool and either apical re-sorting or degradation was argued against in the Discussion. The basis was absence of intracellular detection of the cleaved form but this is not a direct endocytosis assay and controls and assay sensitivity are paramount. The extent of endocytosis should be addressed directly and convincingly.

3) All 3 reviewers had problems with the author's "stop sign" model of how the signal functions. Each expressed this in different ways and each is presented below in the hope that it helps the authors simplify and clarify a model for how the signal might work.

Reviewer 1 "The mechanistic basis for a signal acting as a "stop sign" eludes me. Perhaps this could be clarified. Is this different from a retrieval signal bring escaped protein back to the sorting station? Alternatively, the authors might entertain what I think of as a more conventional view in which the signal (via whatever it binds) is simply strongly positive for apical sorting. In its absence some protein escapes in the basolateral direction and some remains apical (either passively or by some other means)."

Reviewer 2 "The model put forward by the authors to explain their results (Figure 8D) is highly speculative. It suggests that the DxSDE motif functions between Common Recycling Endosomes (CRE) and Apical Recycling Endosomes (ARE), favoring the entrance of Corin into ARE and preventing its entrance into Basolateral Sorting Endosomes (BSE). This is a good possibility since ARE are known to be enriched in Rab11a whereas BSE do not express this protein. However, trafficking studies available to date have not implicated BSE in basolateral targeting so their participation in this route is speculative. Furthermore, whether Rab11a participates at the entrance or at the exit of ARE is not studied in this manuscript (there are no imaging studies showing transit of Corin through ARE). So the possibility exists that Rab11a plays an active role in targeting Corin from ARE to the apical membrane, rather than preventing its entrance into ARE. Whether the basolateral route is emerging before or from the ARE is not clear. Hence, the proposed model should be simpler and simply postulate the involvement of Rab11a positive ARE in promoting apical delivery of proteins displaying the DxSDE motif."

Reviewer 3 "Final paragraph of the Results and the start of the Discussion is confusing. The authors observe that upon expression of DN-RAB11A, a fraction of corin and CD320 are mis-directed to the basolateral surface. On one hand, the authors argue that *RAB11A* is not required for apical expression, but instead acts to prevent these proteins from entering the basolateral trafficking route. They further argue that the DSSDE motif is not an apical sorting determinant, but a stop signal to prevent entry in the basolateral trafficking route. The basis for this argument is unclear, as a protein lacking any sorting information would presumably be delivered in a non-polarized manner: 50% apical and 50% basolateral. Thus, in the absence of *RAB11A* or a functional DSSDE motif, the protein would assume a random distribution, which is what the authors observe. On the other hand, the authors then go on to say that *RAB11A* is involved in endocytic traffic and that apical traffic of these two proteins could be *RAB11A* and signal dependent. It seems that the authors want to rule out a positive role for *RAB11A* because they argue corin and CD320 are not recycling proteins. However, there is a substantial literature that indicates that a subset of newly synthesized apical and basolateral proteins transit endosomes (including *RAB11A*-positive ones) prior to surface delivery. This possibility should be discussed."

---

## [Author Response]

Essential revisions:1) The apical sorting of the constructs is not entirely convincing. There are several issues. Foremost is the lack of quantification of the microscopy-based assay. The authors present single selected images, which is not satisfactory. Additionally, given the lack of resolution of the microscopy in the Z-axis, it is not entirely convincing that the signal being detected is surface-localized rather than, at least partly, intracellular. One reviewer states "the gold standard has been to use cell-surface biotinylation (or related techniques such as cell-surface immunoprecipitation). While transfection efficiency can affect one's ability to perform these studies, the standard for decades has been to make stable cell lines or use viral transduction approaches. These latter approaches could be done using a small subset of the mutants."

We appreciate the reviewers’ concerns. In our figures, each image is representative of the data from at least five experiments. To quantify the results, we analyzed fluorescent intensity along the X-Z axis and calculated the ratio of the intensity on the basolateral membrane *vs*. the total (apical and basolateral) membranes. The quantitative data, which are consistent with the image data, are presented in figure supplements of Figures 2-4, and 7-8. The method for fluorescent signal quantification has been added to the Materials and methods.

As the reviewers suggested, we also did the cell-surface biotin-labeling experiment. Indeed, MDCK cells had low transfection efficiency. We made stable cell lines expressing corin WT and a control mutant (S684A). We detected biotin-labeled corin WT among apical, but not basolateral, membrane proteins, whereas the S684A mutant was detected among apical and basolateral membrane proteins. Together with the data from the trypsin digestion experiment, our results do support the idea that corin is expressed on the apical membrane in MDCK cells. The results from the biotin-labeling experiment are presented in Figure 4—figure supplement 2. The methods are included in the Materials and methods.

2) It is not convincing that the sorting involves direct apical delivery without endocytosis of a basolateral pool of protein. The possibility of endocytosis of a basolateral pool and either apical re-sorting or degradation was argued against in the Discussion. The basis was absence of intracellular detection of the cleaved form but this is not a direct endocytosis assay and controls and assay sensitivity are paramount. The extent of endocytosis should be addressed directly and convincingly.

The reviewers’ point is well taken. Although we showed that the activated corin was not detected intracellularly and that, in the new biotin-labeling experiment, corin was not detected among basolateral membrane proteins, the possibility of endocytosis and recycling cannot be excluded at this time. We have revised the Discussion and pointed out that “Additional studies with more direct and sensitive assays are required to determine if corin and CD320 undergo endocytosis and recycling in MDCK cells”.

3) All 3 reviewers had problems with the author's "stop sign" model of how the signal functions. Each expressed this in different ways and each is presented below in the hope that it helps the authors simplify and clarify a model for how the signal might work.Reviewer 1 "The mechanistic basis for a signal acting as a "stop sign" eludes me. Perhaps this could be clarified. Is this different from a retrieval signal bring escaped protein back to the sorting station? Alternatively, the authors might entertain what I think of as a more conventional view in which the signal (via whatever it binds) is simply strongly positive for apical sorting. In its absence some protein escapes in the basolateral direction and some remains apical (either passively or by some other means)."Reviewer 2 "The model put forward by the authors to explain their results (Figure 8D) is highly speculative. It suggests that the DxSDE motif functions between Common Recycling Endosomes (CRE) and Apical Recycling Endosomes (ARE), favoring the entrance of Corin into ARE and preventing its entrance into Basolateral Sorting Endosomes (BSE). This is a good possibility since ARE are known to be enriched in Rab11a whereas BSE do not express this protein. However, trafficking studies available to date have not implicated BSE in basolateral targeting so their participation in this route is speculative. Furthermore, whether Rab11a participates at the entrance or at the exit of ARE is not studied in this manuscript (there are no imaging studies showing transit of Corin through ARE). So the possibility exists that Rab11a plays an active role in targeting Corin from ARE to the apical membrane, rather than preventing its entrance into ARE. Whether the basolateral route is emerging before or from the ARE is not clear. Hence, the proposed model should be simpler and simply postulate the involvement of Rab11a positive ARE in promoting apical delivery of proteins displaying the DxSDE motif."Reviewer 3 "Final paragraph of the Results and the start of the Discussion is confusing. The authors observe that upon expression of DN-RAB11A, a fraction of corin and CD320 are mis-directed to the basolateral surface. On one hand, the authors argue that RAB11A is not required for apical expression, but instead acts to prevent these proteins from entering the basolateral trafficking route. They further argue that the DSSDE motif is not an apical sorting determinant, but a stop signal to prevent entry in the basolateral trafficking route. The basis for this argument is unclear, as a protein lacking any sorting information would presumably be delivered in a non-polarized manner: 50% apical and 50% basolateral. Thus, in the absence of RAB11A or a functional DSSDE motif, the protein would assume a random distribution, which is what the authors observe. On the other hand, the authors then go on to say that RAB11A is involved in endocytic traffic and that apical traffic of these two proteins could be RAB11A and signal dependent. It seems that the authors want to rule out a positive role for RAB11A because they argue corin and CD320 are not recycling proteins. However, there is a substantial literature that indicates that a subset of newly synthesized apical and basolateral proteins transit endosomes (including RAB11A-positive ones) prior to surface delivery. This possibility should be discussed."

We thank the reviewers for the suggestion. Indeed, our original ‘stop sign’ model and the discussion were speculative and unnecessarily complicated. As the reviewers suggested, we have simplified our model (new Figure 8D) and revised the Results and the Discussion. We have also pointed out that “Further studies are required to understand the molecular basis for the potential Rab11a-dependent recognition mechanism”.